# SEANN: A domain-informed neural network for epidemiological insights

**Jean-Baptiste Guimbaud**[1,2,3], **Marc Plantevit**[4], **Léa Maître**[2], **Rémy Cazabet**[1]*

**1** Universite Claude Bernard Lyon 1, CNRS, INSA Lyon, LIRIS, UMR5205, F-69622 Villeurbanne, France, **2** Barcelona Institute for Global Health (ISGlobal), Barcelona, Spain, **3** Meersens, Lyon, France, **4** EPITA Research Laboratory (LRE), Kremlin-Bicêtre, France

* remy.cazabet@univ-lyon1.fr

## Abstract

In epidemiology, traditional statistical methods such as logistic regression, linear regression, and other parametric models are commonly employed to investigate associations between predictors and health outcomes. However, non-parametric machine learning techniques, such as deep neural networks (DNNs), coupled with explainable AI (XAI) tools, offer new opportunities for this task. Despite their potential, these methods face challenges due to the limited availability of high-quality, high-quantity data in this field. To address these challenges, we introduce SEANN, a novel approach for informed DNNs that leverages a prevalent form of domain-specific knowledge: Pooled Effect Sizes (PES). PESs are commonly found in published Meta-Analysis studies, in different forms, and represent a quantitative form of a scientific consensus. By integrating PES into the training loss, we demonstrate—under controlled simulations—significant improvements in predictive generalization and in the epidemiological plausibility of the learned relationships, relative to a domain-knowledge agnostic neural network.

## 1 Introduction

Historically in epidemiological studies, the impact of environmental health associations was largely studied using a 'one-exposure-one-health-effect' approach [1]. While such targeted approaches are informative, scaling them to the diversity of existing environmental factors is expensive and they can miss complex interactions and unaccounted confounders. The exposome paradigm emerged to address these limitations, proposing to consider the totality of individuals' environmental exposures. This holistic framework allows investigating the combined effects on health from diverse environmental factors, including urban, chemical, lifestyle, and social hazards.

Analyzing such complex mixtures of exposures requires advanced modeling techniques capable of handling high-dimensional data from observational studies [2,3]. While traditional biostatistical methods remain important, recent advancements in

**Data availability statement:** All relevant data are within the manuscript and its Supporting information files.

**Funding:** The research is partially funded by the Meersens Company, financing the salary of the Ph.D. student (first author) (https://meersens.com). The sponsor played no specific role in the research related to this manuscript. There was no additional external funding received for this study. The funders had no role in study design, data collection and analysis, decision to publish, or preparation of the manuscript.

**Competing interests:** The authors have declared that no competing interests exist.

machine learning, particularly Deep Neural Networks (DNNs), offer a powerful alternative for discovering complex patterns in high-dimensional data. Their use, however, poses specific challenges. Their need for large and high-quality datasets and their lack of interpretability are significant obstacles, especially in healthcare, where data can be scarce, noisy, and ethically sensitive.

In dealing with such data, purely data-driven approaches can lead to unsatisfactory results, such as capturing spurious associations. In addition, purely data-driven methods do not comply with known natural laws (e.g., biological pathways). Incorporating additional knowledge to complement the training data has proven its ability to address those issues in various domains [4].

In epidemiology and other cumulative empirical science, scientific knowledge emerges from a consensus among multiple studies, focusing on the same question but in slightly different settings. In the context of the exposome, in which we search for the relation between variables of interest and a target outcome, each study estimates these relations, and these estimations are aggregated across several studies in meta-analyses [5], to derive a more reliable and statistically robust indicator, namely a Pooled Effect Size (PES) [6]. PESs thus represent a quantitative formulation of a scientific consensus. Considered one of the most reliable forms of information in the field [7], PESs have been used to compute literature-only health risk scores (e.g. [8]) and informed risk scores (e.g., [9]) using traditional biostatistical methods (e.g., logistic and linear regression). However, the integration of PESs within machine learning models, particularly DNNs, has not been explored yet, despite their potential to improve the reliability of these models in scarce and noisy data settings.

In this work, we introduce SEANN (Summary Effects Adapted Neural Network), a novel approach designed to integrate prominent forms of PESs, namely Odds Ratios (ORs), Relative Risks (RRs), and Standard Regression Coefficients (SRCs) [10,11] directly into DNNs learning process. By incorporating PESs as a form of scientific prior, SEANN aims to enhance the model's generalizability and ensure that extracted relationships are scientifically plausible. This is achieved through a custom loss function that penalizes deviations from integrated PESs measured through differences in prediction when perturbing the inputs.

SEANN addresses challenges posed by Deep Neural Networks (DNNs) that limit their use in epidemiological contexts, compensating for the limited observational data typically available in exposome-wide studies and improving the generalizability and trustworthiness of computed risk indicators. Additionally, by incorporating knowledge from well-known relationships, SEANN can better characterize those that are less studied.

The paper is structured as follows: in Sect 3, we introduce SEANN; In Sect 4, we perform a series of experiments on synthetic scenarios illustrating the method's benefits in a controlled environment. More specifically, we refer to improved prediction accuracy in noisy contexts and improved reliability of interpretation using XAI. Finally, Sect 5 discusses the significance of those results and concludes.

## 2 Related work

Machine Learning (ML) models and DNNs in particular often rely heavily on both the quantity and the quality of the available training data. In many fields, including healthcare, securing vast and representative datasets poses significant challenges due to ethical, logistical, and technical constraints that would consequently impact the reliability of obtained predictions [12]. Beyond predictive performances, most machine learning procedures do not consider the underlying mechanisms at play (e.g., biological pathways, physical rules, etc) when they learn patterns within the data. As such, they may learn and amplify potential biases, particularly when the data is noisy and incomplete [13].

Informed Machine Learning (IML) addresses these challenges by combining data-driven learning with domain-specific knowledge, leveraging a hybrid approach that can enhance both predictive accuracy and model interpretability [4]. While it is still an emergent stream of research, the number of papers published per year in healthcare alone has approximately doubled every year, starting from 10 published studies in 2018 to 58 in 2021, [14].

The domain-specific knowledge incorporated with IML approaches refers to information not present in the input data. The three most prominent forms of knowledge incorporated using IML in medical applications [14], ranked by decreasing order of importance, are: 1) spatial invariances (e.g., [15]) widely used for image processing, 2) probabilistic relations (e.g., [16]) and 3) knowledge graphs (e.g., [17]). These representations can be integrated into ML models in various ways, such as by modifying the input data, altering the model structure, incorporating knowledge into the loss function, or comparing model outputs with known constraints [4]. Adding penalty terms to the loss function allows the model to balance between fitting the data and adhering to known domain-specific rules. This approach is particularly relevant in cases where learning purely from data may conflict with established domain knowledge. For instance, [18] introduced a Physics-Guided Neural Network (PGNN) incorporating physical rules between water temperature, depth, and density in DNN training ensuring consistency with mechanistic understanding. Similarly, [19] proposed a Domain-Adapted Neural Network (DANN) to integrate domain knowledge for monotonicity, valid ranges and approximation constraints. While DANNs are broadly applicable in fields like physics, engineering, and systems modeling—where knowledge can be expressed as structural constraints or approximate functions—they are less suited to epidemiology, where prior knowledge typically exists as statistical effect-size estimates from previous studies rather than explicit functional forms. Leveraging these effect sizes as a source of prior knowledge offers a natural way to guide learning in this context. Our approach, SEANN, is designed to do exactly this by incorporating PESs into the computation of epidemiological risk scores.

In epidemiological studies, PESs are considered to be among the strongest levels of confidence for a factor's relationship with health at a population level [20]. Despite their significance, there has been limited research on incorporating them into informed machine learning procedures, particularly using deep neural networks (DNNs). To our knowledge, our approach is the first to achieve this. The most similar work, introduced by Neri et al. in 2022 [9], proposed to integrate PESs into a naive Bayes model for the computation of health risk scores, the CArdiovascular LIterature-Based Risk Algorithm (CALIBRA). While their approach can combine input data with literature estimates to learn health relationships, it is limited to naive Bayes models and cannot be directly applied to other types of machine learning procedures.

## 3 Method

This section introduces SEANN. We first define the general setting of integrating PESs as soft constraints via additional terms to the loss function and then detail the implementation of this approach for three types of PESs: standardized regression coefficients, odds ratios, and risk ratios.

Given a set of $p$ observed variables $P$, and a subset $Q \subseteq P$, with $|Q| = q$ of these variables for which we have an effect size estimate value to use as enforced external knowledge, we define **X**, an $n \times p$ input matrix of $n$ observations, and $V$, a vector of $q$ effect size estimates values. Similarly to previous works (e.g., [21,22]), the general principle of our method, described in Eq 1, consists in adding a term to the loss function $\mathcal{L}$ for each meta-heuristic to incorporate.

$$\mathcal{L} = \lambda_0 \mathcal{L}_{pred}(\mathbf{X}, \theta) + \sum_{i=1}^{q} \lambda_i \mathcal{L}_{meta}(\mathbf{X}, \theta, v_i, h_i) \tag{1}$$

Where $\mathcal{L}_{pred}$ is the convex function used for the predictive task (e.g., mean squared error, cross-entropy, etc.) and $\theta$ the parameters of the model. $\mathcal{L}_{meta}$ is the convex loss function used to enforce the desired soft constraints for the neural network, i.e., to enforce the neural network to respect the PESs vector $V$. $\lambda_0$ and $\lambda_i$ are weights pondering the importance of each term, namely the predictive task and the $i^{th}$ constraint. They can be treated as hyperparameters and set to values that optimize the obtained predictive performances. However, for cases where learning plausible relationships, i.e., relationships aligned with known associations, is considered equally or more important than raw predictive power (e.g., imperfect input data, trustworthiness), a different approach should be used to settle the tradeoff between learning from the data (and optimizing performances) or learning associations observed in the literature. We propose choosing weight values proportional to the *confidence* in both the data and the external knowledge. Following a common principle in meta-analyze (e.g., [5]), we express this confidence using the sample size available in each study as well as the sample size available in the training data.

The proposed weighting is calculated as follows: first, we define a confidence score $c_i$ associated with $v_i$ corresponding to the sample size of the $i^{th}$ meta-analysis. Similarly, we define a confidence score $c_0$ for the input data $\mathbf{X}$ composed of $n$ rows and $p$ variables to be computed as $n \times p$. Then, for $c > 1$, we estimate the final $\lambda$ weights using a log scale relative normalization:

$$\lambda_i = \frac{\ln c_i}{\sum_{j=0}^{q} \ln c_j}$$

This scheme ensures that terms associated with small confidence scores have a noticeable impact on the learning process compared to the others and are not entirely ignored.

Depending on the type of PES considered, the loss function $\mathcal{L}_{meta}$ will be implemented differently. As effect estimates in the meta-analysis are typically represented either as OR, RR, or SRC, we express $\mathcal{L}_{meta}$ for those forms below. In all cases, the principle is to generate for each observation a slightly perturbated copy of it with an increment $h$—called the *perturbation*— for each variable in $Q$, to measure the difference between the expected change in the target value according to our PESs and the observed change in our model, and to penalize this difference. A numerical example is provided in Supporting information S1 Text.

### 3.1 Case of a standardized regression coefficient

Let us first consider, for simplicity, a single SRC, called $\beta_i$, that we want to integrate into the training of a DNN. This $\beta_i$ would be either directly extracted in a domain-specific literature study from a uni/multi-variate linear regression model or would summarize several similar effects in a meta-analysis. Considering a multivariate linear regression model defined as:

$$f_{\beta}(\mathbf{Z}) = \beta_0 + \sum_{j=1}^{m_2} \beta_j z_j$$

With $\mathbf{Z} \in \mathbb{R}^{m_1 \times m_2}$ an input matrix and $\beta$, a vector of SRCs, $\beta_i \in \beta$, $1 \leq i \leq m_2$. Then the expected change in the target values according to $\beta_i$ when modifying the corresponding input variable $z_i$ with a perturbation step $h$ is:

$$f_{\beta}(\mathbf{Z}^{z_i+h}) = f_{\beta}(\mathbf{Z}) + \beta_i h$$

Where $\mathbf{Z}^{z_i+h}$ denotes the matrix obtained from the input matrix $\mathbf{Z}$ by perturbing its $i$-th column, denoted as $z_i$, through the addition of a quantity $h$, where $h \in \mathbb{R} \setminus \{0\}$.

To integrate $\beta_i$ within SEANN, we enforce a similar relationship between our model's predicted values $f_\theta(\mathbf{X})$ and predicted values with perturbed inputs $f_\theta(\mathbf{X}^{x_i+h})$ as a soft constraint, i.e., $f_\theta(\mathbf{X}) = f_\theta(\mathbf{X}^{x_i+h}) - \beta_i h$, by penalizing the deviation from this equality.

For a vector $V$ of SRCs derived from the literature, with $v_i \in V$, the $i^{th}$ element of $V$, we propose the following formulation for the training loss term $\mathcal{L}_{meta}$ in the context of SRC integration:

$$\mathcal{L}_{meta}(\mathbf{X}, \theta, v_i, h_i) = \frac{1}{n} \sum_{k=1}^{n} \left( f_\theta(\mathbf{X}_k^{x_i+h}) - v_i h_i - f_\theta(\mathbf{X}_k) \right)^2 \tag{2}$$

Where $\mathbf{X}_k$ denotes the $k^{th}$ row vector of matrix $\mathbf{X}$. $f_\theta$ is the output of the neural network with parameters $\theta$, $n$ the number of data points (i.e., batch size), and $h_i \in \mathbb{R} \setminus \{0\}$ a perturbation parameter. In this case, as SRCs (similar to other PESs) are constant regardless of input data $\mathbf{Z}$, we can theoretically use any value other than 0. For simplicity, we use $h_i = 1$ for every SRCs to integrate. In a hypothetical, more general case of a constraint to integrate as a function of $\mathbf{X}$, $h$ would be taken as the smallest possible.

### 3.2 Case of an odds-ratio

The approach we proposed in this section, while mathematically correct, can suffer from numerical instability during the training (cf., Eq 3). In paper 3, we addressed this issue by generalizing Eq 2 instead.

Similar to Sect 3.1, let us consider a single OR, referred to as $(OR_i = e^{\beta_i})$, that we want to integrate into the training process of a DNN. This OR would be extracted from logistic regression models in a meta-analysis. Considering a multivariate logistic regression model defined as:

$$p_\beta(\mathbf{Z}) = \frac{1}{1 + e^{-\left(\beta_0 + \sum_{j=1}^{m_2} \beta_j z_j\right)}}$$

With $\mathbf{Z} \in \mathbb{R}^{m_1 \times m_2}$ an input matrix and $\beta$, a vector of $m_2$ log-odds coefficients, $\beta_i \in \beta$, $1 \leq i \leq m_2$. We can express the change in $Logit(p_\beta)$ when modifying the input variable $z_i$ associated with $\beta_i$ with a perturbation step $h$. This difference is independent of other variables in $\mathbf{Z}$.

$$\log\left(\frac{p_\beta(\mathbf{Z}^{z_i-h})}{1 - p_\beta(\mathbf{Z}^{z_i-h})}\right) - \log\left(\frac{p_\beta(\mathbf{Z})}{1 - p_\beta(\mathbf{Z})}\right) = -\beta_i h$$

Thus, we can express the corresponding relationship between the predicted values on inputs $\mathbf{Z}$ with and without modifying the corresponding input variable $z_i$ with a perturbation step $h$:

$$p_\beta(\mathbf{Z}) = \frac{e^{\beta_i h} p_\beta(\mathbf{Z}^{z_i-h})}{e^{\beta_i h} p_\beta(\mathbf{Z}^{z_i-h}) - p_\beta(\mathbf{Z}^{z_i-h}) + 1}$$

For a vector $V$ of log-odds coefficients derived from the literature, with $v_i \in V$, the $i^{th}$ element of $V$, we propose the following formulation for the training loss term $\mathcal{L}_{meta}$ in the context of OR integration:

$$\mathcal{L}_{meta}(\mathbf{X}, \theta, v_i, h_i) = \frac{1}{n} \sum_{k=1}^{n} \left( \frac{e^{v_i h_i} p_\theta(\mathbf{X}_k^{x_i-h})}{e^{v_i h_i} p_\theta(\mathbf{X}_k^{x_i-h}) - p_\theta(\mathbf{X}_k^{x_i-h}) + 1} - p_\theta(\mathbf{X}_k) \right)^2 \tag{3}$$

Where $p_\theta$ is the probability given by the neural network with parameters $\theta$, $n$ the number of data points (i.e., the batch size) and $h \in \mathbb{R} \setminus \{0\}$ a perturbation parameter. Similar to Sect 3.1, as a given OR is constant for every corresponding $z$, $h$ can theoretically take any values other than 0. However, within SEANN, we fix $h$ to keep the quantities within the exponential terms small and enhance numerical stability during the learning process. We define:

$$h = \begin{cases} 1 & \text{if } v_i = 0, \\ \dfrac{1}{v_i} & \text{otherwise.} \end{cases}$$

### 3.3 Case of a risk ratio

Following the same principle, let's define the integration of a single PES encoded as a risk ratio. Considering a negative binomial regression model defined as:

$$\log\left(\mu_\beta(\mathbf{Z})\right) = \beta_0 + \sum_{j=1}^{m_2} \beta_j z_j$$

With $\mathbf{Z} \in \mathbb{R}^{m_1 \times m_2}$ an input matrix and $\beta$, a vector of log-estimates, $\beta_i \in \beta$, $1 \le i \le m_2$. Then the expected change in the target values according to $\beta_i$ when modifying the corresponding input variable $z_i$ with a perturbation step $h$ is:

$$\mu_\beta(\mathbf{Z}^{z_i+h}) = e^{\beta_i h} \mu_\beta(\mathbf{Z})$$

Where $\mathbf{Z}^{z_i+h}$ denotes the matrix obtained from the input matrix $\mathbf{Z}$ by perturbing its $i$-th column, denoted as $z_i$, through the addition of a quantity $h$, where $h \in \mathbb{R} \setminus \{0\}$.

To integrate $\beta_i$ within SEANN, we enforce a similar relationship between our model's predicted values $f_\theta(\mathbf{X})$ and predicted values with perturbed inputs $f_\theta(\mathbf{X}^{x_i+h})$ as a soft constraint, i.e., $f_\theta(\mathbf{X}) = f_\theta(\mathbf{X}^{x_i+h})e^{-\beta_i h}$, by penalizing the deviation from this equality.

For a vector $V$ of log-estimates derived from the literature, with $v_i \in V$, the $i^{th}$ element of $V$, we propose the following formulation for the training loss term $\mathcal{L}_{meta}$ in the context of RR integration:

$$\mathcal{L}_{meta}(\mathbf{X}, \theta, v_i, h_i) = \frac{1}{n} \sum_{k=1}^{n} \left( e^{-v_i h_i} f_\theta(\mathbf{X}_k^{x_i+h}) - f_\theta(\mathbf{X}_k) \right)^2 \tag{4}$$

Where $\mathbf{X}_k$ denotes the $k^{th}$ row vector of matrix $\mathbf{X}$. $f_\theta$ is the output of the neural network with parameters $\theta$, $n$ the number of data points (i.e., batch size), and $h_i \in \mathbb{R} \setminus \{0\}$ a perturbation parameter. Similar to Sect 3.2, in order to keep the quantities within the exponential term small and enhance numerical stability, we recommend using:

$$h = \begin{cases} 1 & \text{if } v_i = 0, \\ -\dfrac{1}{v_i} & \text{otherwise.} \end{cases}$$

To demonstrate the potential of the approach, we rely on synthetic data that emulate different scenarios. In each experiment, we compare two DNNs, identical in all aspects but with the inclusion of our modified loss and external knowledge. For the sake of simplicity, we use and compare basic multilayer perceptrons. The approach can be directly usable with more complex feedforward neural architectures (e.g., convolutional networks, residual networks), and the benefits highlighted in this study should apply to other neural configurations. In the following, we call SEANN the model that implements our approach and *the agnostic DNN* the reference.

## 4 Experimental validation

### 4.1 Data scenario

To illustrate the relevance in real applications, we introduce an intuitive fictional example composed of 1) a target variable $y$ representing the risk of developing a disease or the strength of symptoms and 2) several variables contributing to the outcome $y$ according to a dose-response relationship. To make the experiment more intuitive, we take a well-known confounding factor in environmental epidemiology: Reasonable fish intake tends to decrease health-related risks (e.g., cardio-vascular disease and cognitive decline), while mercury intake increases them. However, fish intake is an important driver of mercury exposure. Thus, we define two correlated variables, *mercury* ($x_1$) and *fish intake* ($x_2$), having an opposite effect on the target variable. The correlation between $x_1$ and $x_2$ is designed to emulate a confounding effect [23]. We also define two additional variables, namely *perceived stress* ($x_3$) and *body mass index* (i.e., BMI, $x_4$) uncorrelated with variables $x_1$ and $x_2$ but affecting $y$. $x_3$ is linear and positively correlated with the outcome, while $x_4$ has a nonlinear effect.

In this simple scenario, we perform different experiments in which we test the benefits of incorporating PESs into eligible variables (i.e., $x_1$, $x_2$ and $x_3$). We are interested not only in the networks' predictive performances, but also in their ability to capture and restitute the input-output relationships that we encoded in the data. The experiments focus on SRCs and ORs, but we could use RRs in a similar manner.

**4.1.1 Standardized regression coefficients.** For the case where PESs are encoded as SRCs, we generate an input matrix $\mathbf{X}$ by sampling $m = 1000$ values from a multivariate Gaussian with mean 0 and covariance matrix $\left(\begin{smallmatrix} 1 & 0.8 & 0 & 0 \\ 0.8 & 1 & 0 & 0 \\ 0 & 0 & 1 & 0 \\ 0 & 0 & 0 & 1 \end{smallmatrix}\right)$. Target vector $Y$ is generated from the additive function described in Eq 5 with $\beta_0 = 1$, $\beta_1 = 1$, $\beta_2 = -2$, $\beta_3 = 5$ and $\beta_4 = 10$.

$$Y(\mathbf{X}) = \beta_0 + \beta_1 \times x_1 + \beta_2 \times x_2 + \beta_3 \times x_3 + \beta_4 \times \cos(x_4) \tag{5}$$

**4.1.2 Odds-ratios.** Similar to the linear case, we generate a data matrix $\mathbf{X}$ with three variables, namely $x_1$, $x_2$, and $x_3$, corresponding to mercury, fish intake, and perceived stress, respectively, to predict an outcome (i.e., a risk to develop a disease). $\mathbf{X}$ was generated by sampling $m = 1000$ values from a multivariate Gaussian with mean 0 and covariance matrix $\left(\begin{smallmatrix} 1 & 0.8 & 0 \\ 0.8 & 1 & 0 \\ 0 & 0 & 1 \end{smallmatrix}\right)$. Target vector $Y$ was generated from the function described in Eq 6 with $\beta_0 = 0$, $\beta_1 = 1$, $\beta_2 = -2$, $\beta_3 = 5$.

$$Y(\mathbf{X}) = \frac{1}{1 + e^{-\beta_0 - \beta_1 \times x_1 - \beta_2 \times x_2 - \beta_3 \times x_3}} \tag{6}$$

**4.1.3 Experimental design.** We use a fully connected neural network (NN) with a single hidden layer for both SEANN and the agnostic model. Both NNs were implemented using Pytorch and trained with a batch size of 64 and a maximum number of epochs of 1000. Parameter optimization was achieved using Adam [24]. We standardize and split data into training (n=600), validation (n=200), and test (n=200) datasets. To reduce overfitting, we use early stopping (with patience 10) on the validation set.

**4.1.4 Evaluation.** To evaluate the correctness of extracted relationships, we propose a score, called $\Delta Shap$, to compute the distance between two dose-response relationships represented with Shapley values [25]. $\Delta Shap$ is defined by the mean absolute error (MAE) calculated across Shapley values for a given marginal relationship that must be computed using the same background dataset. We can use it to compare the distance between a neural network-extracted relationship and a relationship admitted in the literature or, in this work, as we use synthetic data, to compare the distance between a neural network-extracted relationship and the true predictor-outcome relationship. The smaller the distance, the more we would consider a relationship to be scientifically plausible or in line with the true effect. In this work, we use the generative functions (i.e., Eq 5 and Eq 6) to compute Shapley values representing the reference relationships for $\Delta Shap$. Shapley values are approximated using the SHAP library [26] and systematically computed on the test sets.

To evaluate the performance of models, we use the coefficient of determination ($R^2$) score for regression tasks and the receiver operating characteristic curve's ($ROC$) area under the curve ($AUC$) for binary classification tasks.

### 4.2 Experiment 1

In this experiment, we illustrate that SEANN can leverage external expert information to mitigate the poor quality of the data. To simulate data imperfection, we gradually increase the level of noise on all variables (i.e., $x_1$, $x_2$, $x_3$, $x_4$) and check that while the performance of the agnostic NN deteriorates quickly, our informed NN can retain most of it. Information was degraded differently for linear coefficients and odds ratios to illustrate two common scenarios. In the linear case, we added Gaussian noise to all input variables, with a mean of 0 and an increasing standard deviation. For odds ratios, missing values were generated completely at random with increasing proportions and imputed using a simple mean imputation. External knowledge was integrated in addition to the training data for every eligible predictor (i.e., $\beta_1$, …, $\beta_3$ for $x_1$, …, $x_3$ respectively).

Better performances were obtained with SEANN both for the predictive task and the explainability of constrained relationships measured with $\Delta Shap$. No significant gains were observed for the nonlinear unconstrained variable (BMI). The results are displayed in Fig 1 for the linear coefficients and summarized in Table 1 for the odds ratios.

The results indicate that when the PESs encode the correct relationships between the input variables and the target outcome, the predictive performances obtained while training on imperfect data are more stable with SEANN and the captured relationships are more scientifically credible. From a health research perspective, this means that when epidemiological data are noisy or incomplete—a common situation in cohort studies—SEANN can still recover relationships that are consistent with established knowledge. For example, while a standard NN tends to distort exposure effects as data quality deteriorates, SEANN preserves plausible dose–response patterns for mercury, fish intake, and stress by leveraging literature-informed constraints. In practice, this reduces the likelihood of drawing misleading conclusions about exposure–outcome relationships and improves the ability to retrieve meaningful effects under imperfect data conditions.

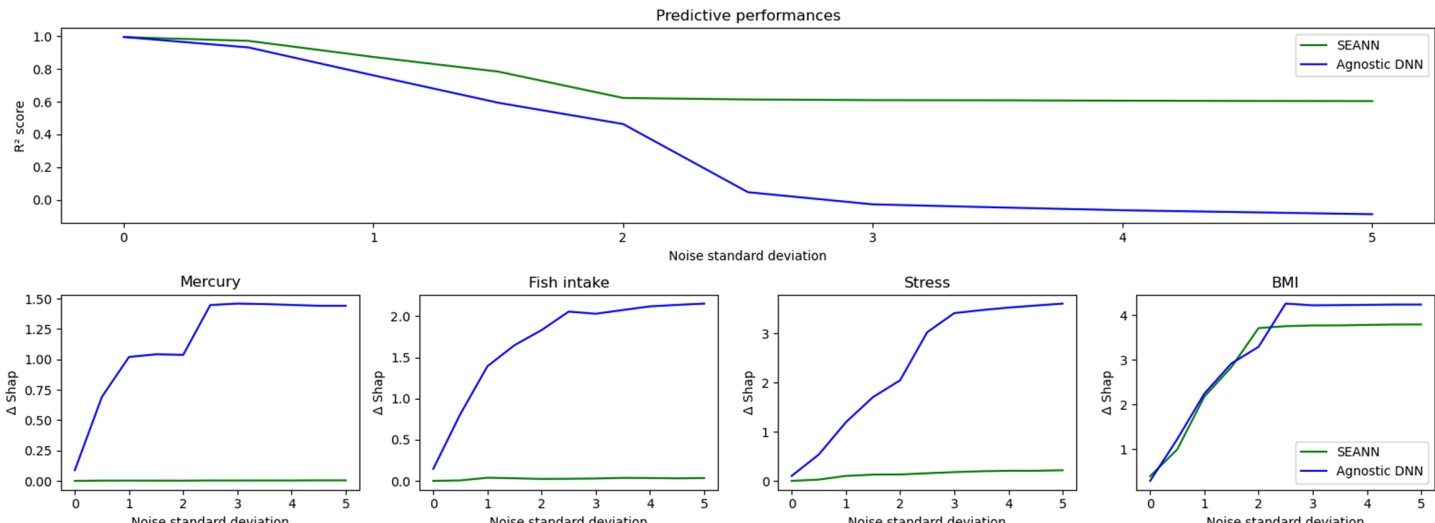

**Fig 1**. **Performance comparison of SEANN and an agnostic DNN under increasing noise (Experiment 1).** The x-axis shows the standard deviation of Gaussian noise added to all predictors. The y-axes report (top panel) predictive performance ($R^2$) and (bottom panel) the alignment of extracted dose–response relationships with the ground truth, quantified by ΔShap (lower is better). Across noise levels, SEANN maintains higher predictive performance and more accurate relationships than the agnostic DNN.

**Table 1**. **Comparison of Performances depending on the proportion of imputed missing values in training and validation sets (experiment 1).** ROC AUC is a measure of predictive performances, whereas Sum Δ Shap summarizes the quality of captured relationships across all predictors. As noise level increases, SEANN maintains higher predictive performance and more accurate relationships than the agnostic DNN.

| Percent missing | Agnostic DNN | | SEANN | |
|---|---|---|---|---|
| | ROC AUC | Sum Δ Shap | ROC AUC | Sum Δ Shap |
| 0 | 0.999 | 0.083 | 0.998 | 0.137 |
| 25 | 0.930 | 0.262 | 0.975 | 0.176 |
| 50 | 0.915 | 0.318 | 0.95 | 0.204 |

## 4.3 Experiment 2

In this experiment, we focus on the quality of the relationships captured when external knowledge is integrated for a single predictor. The objective is to show that variables that do not benefit directly from external knowledge may nevertheless be better captured, thanks to corrections brought to the other variables. Similarly to the previous experiment, Gaussian noise was added to a single variable (i.e., fish intake, $x_2$) with mean 0 and standard deviation 0.75, and 1.5 for the SRCs and ORs respectively, in both training and validation sets.

Fig 2 show results with PESs encoded as SRCs. The most significant gain was observed for fish intake, the variable with integrated external knowledge. $\Delta Shap$ (see definition in Sect 4.1.4) measured for this variable was 1.11 with the agnostic DNN and 0.04 with SEANN. A significant gain was also observed for mercury, the variable correlated with the informed one (1.02 for agnostic DNN to 0.53 for SEANN). Finally, no significant performance gains were observed for the remaining variables, i.e., those uncorrelated with the informed one (0.99 to 1.04 for perceived stress, 1.74 to 1.64 for BMI, with agnostic DNN and SEANN, respectively).

The results show that not only SEANN better captures relationships for features with corresponding external knowledge but also for non-informed features that are correlated with those externally informed. From a health research perspective, these results illustrate that SEANN not only improves the interpretation of exposures with prior knowledge but

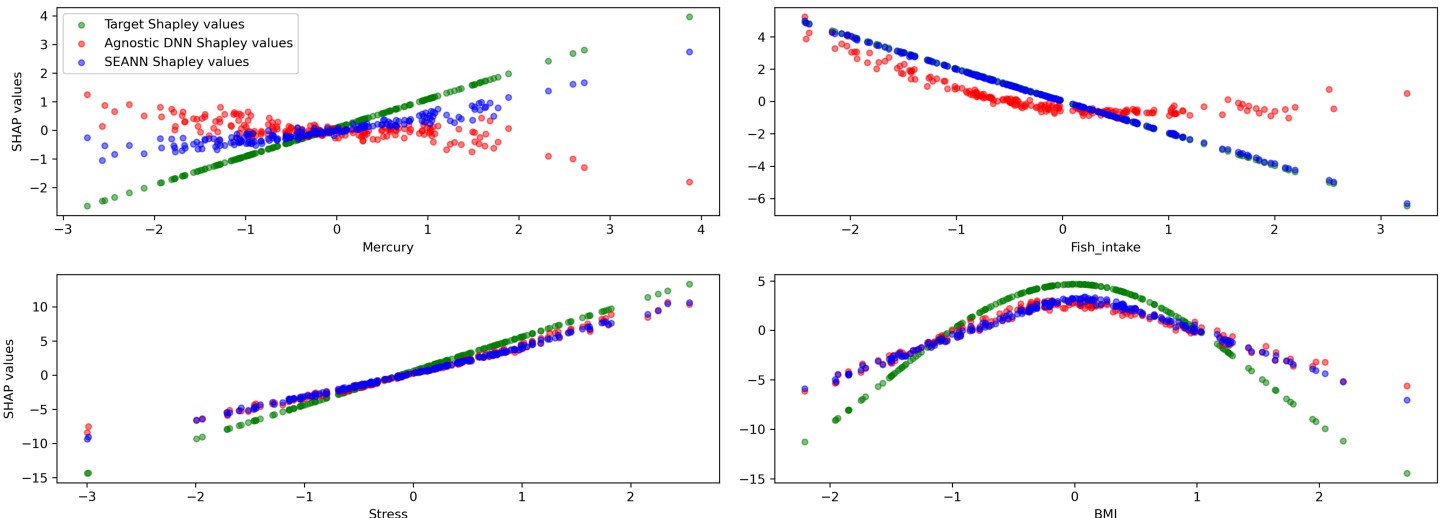

**Fig 2**. **Comparison of extracted relationships (Shapley values) between the agnostic DNN and SEANN (experiment 2).** Only the beta coefficient for fish intake was added as external knowledge. SEANN not only recovers the expected protective effect of fish intake (close alignment with external knowledge), but also improves the capture of the correlated harmful effect of mercury exposure. By contrast, the agnostic DNN produces distorted or attenuated associations under the same noisy conditions.

also enhances the reliability of correlated predictors, helping disentangle complex mixtures where exposures co-occur. In practice, this means that known dose–response patterns (e.g., for fish intake) can guide the recovery of related effects (e.g., mercury), reducing bias from collinearity and enabling more trustworthy interpretation of intertwined environmental exposures.

A similar scenario is observed for odds ratios, with $\Delta Shap$ down from 0.087 with the agnostic DNN to 0.050 with SEANN for mercury.

## 4.4 Experiment 3

In a last experiment, we simulate a setting where a confounding variable is missing from the data. A confounding variable is a predictor impacting both the outcome to predict and other predictor(s) of interest. In numerous contexts, including health science, it is challenging to collect all relevant variables to predict an outcome (and study their effects), and we can expect to have unseen confounders.

We train both NNs with a missing variable (fish intake) and compare both the predictive performance and the quality of the extracted relationships on the test set. With SEANN, we integrate external knowledge for mercury alone (i.e., the variable correlated with the missing predictor), and we duplicate the mercury variable in both training and validation datasets, the copy being an unconstrained variable. The objective of this duplication is that 1) The constrained version captures what is known by external knowledge, and 2) The unconstrained version captures what comes from the unseen confounder.

We observe (Fig 3) that without the constraint, mercury is incorrectly captured, with a $\Delta Shap$ of 1.32. In this case, interpreting the Shapley values directly could lead to the misleading conclusion that mercury has a protective effect. On the contrary, with SEANN, the constraint allows the capture of the correct relation ($\Delta Shap$=0.013). Additionally, SEANN was able to capture part of the association with the missing variable (fish intake) using duplicated input data of mercury ($\Delta Shap$: 0.43). Minor improvements were also observed for other variables. Results show that SEANN can be used to better disentangle individual effects while estimating the effect of unknown confounding factors.

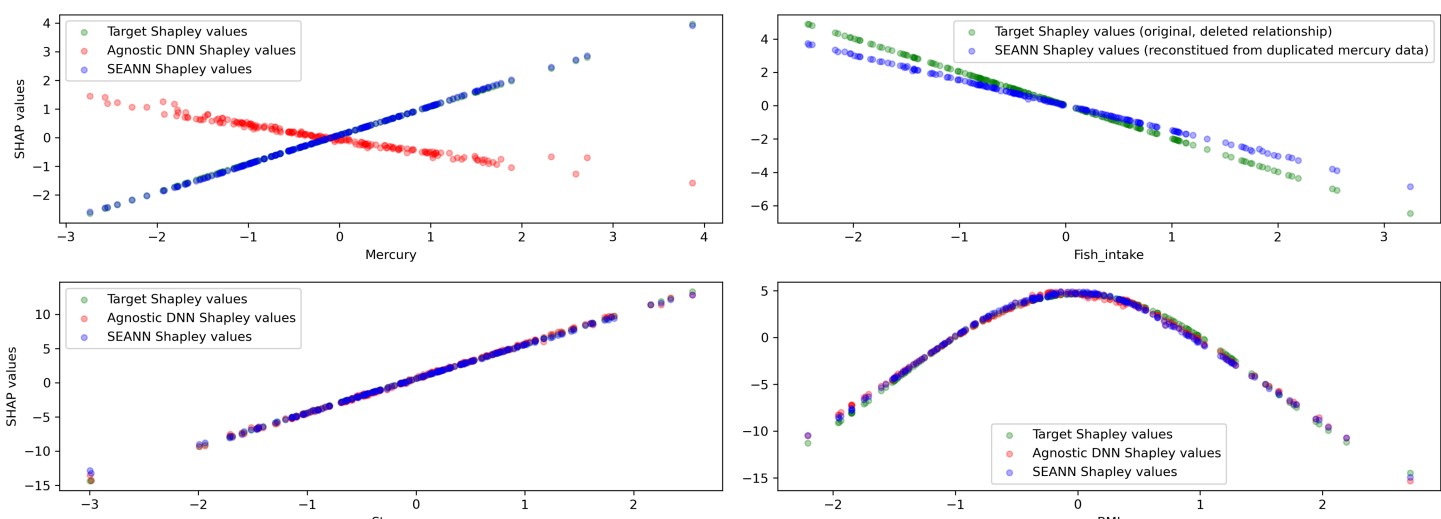

**Fig 3**. **Comparison of extracted relationships (Shapley values) between the agnostic DNN and SEANN in the presence of a missing confounder (fish intake, Experiment 3).** Without external knowledge, the agnostic DNN misinterprets mercury as protective, leading to misleading conclusions. SEANN, by constraining the mercury effect with PES information, correctly recovers its harmful association and partially captures the influence of the missing fish intake variable through the duplicated feature.

For odds ratios, the results are similar. SEANN better captured the mercury predictor, with $\Delta Shap$=0.128 for the agnostic DNN and 0.048 for SEANN. SEANN was also able to capture part of the association with the missing variable (i.e., fish intake) using the duplicated input data ($\Delta Shap$= 0.056).

## 5 Discussion

In this paper, we propose a method to integrate the wealth of knowledge available in the scientific literature encoded as pooled effect sizes (PESs). While these representations are simple estimates unable to express complex relationships, they are easily understandable, can be aggregated across multiple studies, and are widely used in epidemiology. By integrating these traditional statistical measures into the deep learning process, our approach offers a tool to capture complex nonlinear relationships from data while leveraging simpler but well-established knowledge. Our findings complement existing informed-ML approaches that encode domain knowledge as structural or mechanistic soft constraints (e.g., physics-guided networks or monotonic/range constraints implemented via loss regularization). In contrast, while prior work has injected effect-size estimates into simpler models such as naïve Bayes (e.g., CALIBRA), to our knowledge no method has integrated pooled effect sizes directly into neural network training, despite their prominence in epidemiological research.

Our experimental protocol demonstrates that, compared with an agnostic DNNs with similar architecture, our approach offers two main benefits. First, improved generalization to unseen data: when PESs encode relevant information that is weak or absent in the observed data, SEANN's predictive performance degrades more slowly under noise and missingness, yielding higher out-of-sample accuracy. Second, improved alignment of extracted relationships with external knowledge: SEANN preserves epidemiologically plausible dose–response patterns for both informed and uninformed variables and can be used to better disentangle individual input-output relationships in the presence of collinearity. For real-world epidemiological studies, these benefits imply more reliable risk predictions across different cohorts, populations or sites and more stable, policy-relevant, effect estimates despite measurement error, limited sample sizes, and partial confounding.

Beyond the controlled synthetic experiments presented in this study, SEANN has already been applied to real-world epidemiological data. In a recent study on hypertension risk in European adults [27], we integrated SEANN into the construction of environmental risk scores using observational cohort data (from the GCAT project [28]) combining diverse environmental, sociodemographic, and clinical exposures. In this high-dimensional and partially confounded setting, SEANN improved the quality of captured relationships compared with agnostic neural networks, confirming its practical utility for epidemiological modeling. Building on our work on the early-life exposome in HELIX [29,30], future work will extend SEANN to life-course risk modeling and to larger multi-site cohorts, with the goal of enhancing environmental clinical risk scores.

## Supporting information

**S1 Text**.
(TEX)

## Author contributions

**Conceptualization:** Jean-Baptiste Guimbaud, Lea Maitre, Marc Plantevit, Remy Cazabet.

**Funding acquisition:** Marc Plantevit, Remy Cazabet.

**Investigation:** Jean-Baptiste Guimbaud, Marc Plantevit, Remy Cazabet.

**Methodology:** Jean-Baptiste Guimbaud, Remy Cazabet.

**Project administration:** Lea Maitre.

**Software:** Jean-Baptiste Guimbaud.

**Supervision:** Lea Maitre, Remy Cazabet.

**Writing – original draft:** Jean-Baptiste Guimbaud, Remy Cazabet.

**Writing – review & editing:** Jean-Baptiste Guimbaud, Lea Maitre, Marc Plantevit, Remy Cazabet.

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
