## [Decision Letter · Decision Letter 0]

8 Jul 2025

PONE-D-25-02109SEANN: A Domain-Informed Neural Network for Epidemiological InsightsPLOS ONE

Dear Dr. Cazabet,

Thank you for submitting your manuscript to PLOS ONE. After careful consideration, we feel that it has merit but does not fully meet PLOS ONE’s publication criteria as it currently stands. Therefore, we invite you to submit a revised version of the manuscript that addresses the points raised during the review process.

We look forward to receiving your revised manuscript.

Kind regards,

Vijay Kumar

Academic Editor

PLOS ONE

Journal Requirements:

“The research is partially funded by the Meersens Company, financing the salary of the Ph.D student (first author) (https://meersens.com). The sponsor played no specific role in the research related to this manuscript.”

“The research is partially funded by the Meersens Company, financing the salary of the Ph.D student (first author) (https://meersens.com). The sponsor played no specific role in the research related to this manuscript.”

5. We note that your Data Availability Statement is currently as follows: All relevant data are within the manuscript and its Supporting Information files.

Reviewers' comments:

Reviewer's Responses to Questions

**Comments to the Author**

1. Is the manuscript technically sound, and do the data support the conclusions?

Reviewer #1: Yes

Reviewer #2: Yes

Reviewer #3: No

2. Has the statistical analysis been performed appropriately and rigorously?

Reviewer #1: Yes

Reviewer #2: Yes

Reviewer #3: No

3. Have the authors made all data underlying the findings in their manuscript fully available?

Reviewer #1: Yes

Reviewer #2: No

Reviewer #3: Yes

4. Is the manuscript presented in an intelligible fashion and written in standard English?

Reviewer #1: Yes

Reviewer #2: Yes

Reviewer #3: Yes

5. Review Comments to the Author

Reviewer #1: General Comments:

The manuscript presents a novel and relevant approach by introducing SEANN, a domain-informed neural network model that leverages pooled effect sizes (PES) to enhance prediction and interpretation in epidemiological datasets. The work is timely and contributes to the growing intersection of machine learning and health sciences, especially in low-data settings. The manuscript is written in good English; however, it requires minor revisions before it can be accepted for publication.

Specific Comments:

1. The manuscript, while clearly written, lacks consistent formatting throughout. Please ensure that headings, equations, and paragraphs follow the journal’s style guide. Several in-line equations are not properly spaced and disrupt the flow of reading.

2. Authors should check for the typographical errors such as in the Introduction section: “Odd Rations (ORs)” should be revised to “Odds Ratios (ORs).”

3. Authors should check for citing relevant citations in the text for e.g. On Page 5, the following line is missing a relevant citation:

"For a vector V of SRCs derived from the literature, with vi ∈ V, the ith element of V, the training loss term Lmeta is defined as..."

Cite appropriate literature to support the use of the custom loss formulation.

4. While the analysis appears comprehensive, the discussion lacks sufficient depth. It is recommended that the authors elaborate more on how their results compare with prior methods and explain the implications of improved generalizability in real-world epidemiological studies.

Reviewer #2: 1.This manuscript introduces a novel approach, SEANN (Summary Effects Adapted Neural Network), which integrates pooled effect sizes (PESs) such as Odds ratios, Risk ratios, and standardized regression coefficients directly into deep neural network training. This is a unique and timely contribution to informed machine learning, particularly relevant to fields like epidemiology, where data is often limited or noisy. SEANN helps improve both predictive performance and interpretability by embedding prior scientific knowledge into model training.

2.The methodology is technically solid and clearly presented. The paper explains how PESs are embedded via custom loss functions and how weighting is derived based on confidence (sample size). The treatment of three common types of PESs is thorough. However, while the math is sound, the use of complex symbols and notations (Example: perturbation steps, arrow symbols) might challenge readers outside machine learning. Adding a simpler example with numeric inputs could help clarify the non-expert audiences.

3.The experiments are well-structured and simulate realistic epidemiological challenges noise, missing data, and confounding. The use of synthetic data is reasonable for isolating the effects of the method. SEANN consistently outperforms standard DNNs in terms of both prediction and interpretability, especially when data is imperfect. The use of the SHAP metric to assess alignment with true relationships is particularly valuable. However, the study would benefit from including comparisons with other informed machine learning models, like PGNN or DANN, or at least discussing their relative merits.

4.SEANN enhances the interpretability of model predictions, aligning well with real-world needs in health research. For example, it corrects misleading associations when variables are confounded or missing. While the figures and quantitative results are strong, some brief narrative explanations of what those corrections mean in practice, especially for health researchers, would make the work more engaging and accessible.

5.This is a strong and original piece of work. To improve it further, the authors should consider: Including a real-world dataset example.

Reviewer #3: While the paper presents a creative idea — integrating domain knowledge in the form of pooled effect sizes into a neural network via a modified loss function — the manuscript ultimately lacks sufficient scientific rigor and practical relevance to warrant publication in PLOS ONE. My detailed concerns are as follows:

1. Lack of Real Data: The manuscript claims to improve epidemiological modeling, yet it contains no experiments on real-world epidemiological datasets. Synthetic toy examples (e.g., fish, mercury, stress, BMI) are insufficient to demonstrate the utility of the approach. Without validation on observational datasets from real health studies, the results cannot be trusted or generalized.

2. Overclaiming Practical Use: The term “epidemiological insights” in the title and abstract is misleading. No insights about disease risk, exposure, or population health are produced in this study. The method is demonstrated only in artificial settings, using pre-defined effects, which defeats the purpose of “insight” discovery.

3. Weak Baselines and Limited Evaluation: The authors compare SEANN only to an “agnostic DNN” — essentially a DNN without the PES-informed loss — and omit any comparison to:

- Classical epidemiological methods (e.g., regularized logistic regression with prior constraints),

- Knowledge-regularized models in causal inference,

- Other informed machine learning models in epidemiology (e.g., those incorporating DAGs or monotonic constraints).

The absence of meaningful baselines diminishes the credibility of the claimed improvements.

4. Unconvincing Justification of PES Integration: The proposed method modifies the loss function using hand-selected perturbation values (e.g., h=1 or h=1/vi), but these choices are not theoretically or empirically justified. There is also no sensitivity analysis regarding how these perturbations affect training stability or generalization. Furthermore, integrating PES values — which are often heterogeneous and context-dependent — as “hard-coded” truth in synthetic experiments does not reflect how they behave in actual epidemiological studies.

5. Limited Contribution to Machine Learning or Epidemiology: While the paper claims novelty in the integration of PESs into neural networks, the broader field of informed ML has explored similar loss regularization strategies with more robust theory and applications. This work does not meaningfully advance the methodological or epidemiological literature beyond existing approaches.

6. Figures and Tables Are Minimal and Not Informative Enough: The visualizations do not adequately explain how SEANN outperforms baselines. Key graphs (e.g., SHAP plots) are not contextualized with respect to meaningful health variables, and only small performance differences are observed in some metrics.

6. PLOS authors have the option to publish the peer review history of their article (what does this mean?). If published, this will include your full peer review and any attached files.

Reviewer #1: No

Reviewer #2: No

Reviewer #3: No

---

## [Author Response · Author response to Decision Letter 1]

21 Oct 2025

We uploaded a PDF document with detailed answers to reviewers and editor.

---

## [Editor Report · Decision Letter 1]

23 Nov 2025

SEANN: A Domain-Informed Neural Network for Epidemiological Insights

PONE-D-25-02109R1

Dear Dr. Cazabet,

We’re pleased to inform you that your manuscript has been judged scientifically suitable for publication and will be formally accepted for publication once it meets all outstanding technical requirements.

Kind regards,

Vijay Kumar

Academic Editor

PLOS ONE

Additional Editor Comments (optional):

As I have considered that the authors appropriately addressed all the comments from the reviewers, the manuscript can be accepted.
---

## [Editor Report · Acceptance letter]

PONE-D-25-02109R1

PLOS ONE

Dear Dr. Cazabet,

I'm pleased to inform you that your manuscript has been deemed suitable for publication in PLOS ONE. Congratulations! Your manuscript is now being handed over to our production team.

Kind regards,

on behalf of

Dr. Vijay Kumar

Academic Editor

PLOS ONE